# On the causes of gene-body methylation variation in *Arabidopsis thaliana*

**Rahul Pisupati[1,2], Viktoria Nizhynska[1], Almudena Mollá Morales[1], Magnus Nordborg[1]** *

**1** Gregor Mendel Institute, Austrian Academy of Sciences, Vienna BioCenter (VBC), Vienna, Austria,
**2** Vienna Graduate School of Population Genetics, Institut für Populationsgenetik, Vetmeduni, Vienna, Austria

* magnus.nordborg@gmi.oeaw.ac.at

**Data Availability Statement:** The raw sequencing data and the methylation calls are uploaded to the NCBI GEO database under GSE215839. All scripts and relevant intermediate data files are available on

## Abstract

Gene-body methylation (gbM) refers to sparse CG methylation of coding regions, which is especially prominent in evolutionarily conserved house-keeping genes. It is found in both plants and animals, but is directly and stably (epigenetically) inherited over multiple generations in the former. Studies in *Arabidopsis thaliana* have demonstrated that plants originating from different parts of the world exhibit genome-wide differences in gbM, which could reflect direct selection on gbM, but which could also reflect an epigenetic memory of ancestral genetic and/or environmental factors.

Here we look for evidence of such factors in F2 plants resulting from a cross between a southern Swedish line with low gbM and a northern Swedish line with high gbM, grown at two different temperatures. Using bisulfite-sequencing data with nucleotide-level resolution on hundreds of individuals, we confirm that CG sites are either methylated (nearly 100% methylation across sampled cells) or unmethylated (approximately 0% methylation across sampled cells), and show that the higher level of gbM in the northern line is due to more sites being methylated. Furthermore, methylation variants almost always show Mendelian segregation, consistent with their being directly and stably inherited through meiosis.

To explore how the differences between the parental lines could have arisen, we focused on somatic deviations from the inherited state, distinguishing between gains (relative to the inherited 0% methylation) and losses (relative to the inherited 100% methylation) at each site in the F2 generation. We demonstrate that deviations predominantly affect sites that differ between the parental lines, consistent with these sites being more mutable. Gains and losses behave very differently in terms of the genomic distribution, and are influenced by the local chromatin state. We find clear evidence for different trans-acting genetic polymorphism affecting gains and losses, with those affecting gains showing strong environmental interactions (G×E). Direct effects of the environment were minimal.

In conclusion, we show that genetic and environmental factors can change gbM at a cellular level, and hypothesize that these factors can also lead to transgenerational differences between individuals via the inclusion of such changes in the zygote. If true, this could explain genographic pattern of gbM with selection, and would cast doubt on estimates of epimutation rates from inbred lines in constant environments.

Github (http://github.com/Gregor-Mendel-Institute/pisupati-gbm-paper-2022.git).

**Funding:** This work was supported in part by European Research Council (ERC) Advanced Grant 789037 to MN. The funders had no role in study design, data collection and analysis, decision to publish, or preparation of the manuscript.

**Competing interests:** The authors have declared that no competing interests exist.

## Author summary

Gene-body methylation, the sparse CG methylation that is associated with house-keeping genes, is found in both plants and animals, but can be directly inherited in the former. Recently, we discovered that *Arabidopsis thaliana* originating from different geographic regions exhibit different patterns of gbM, which could be due to direct selection on gbM, but could also reflect a transgenerational memory of genetic or environmental factors. Here we look for evidence of such factors using a genetic cross between two natural inbred lines: one with high, and one with low gbM. We confirm that methylation states are stably inherited, but also see large somatic deviations from the inherited state, in particular at sites that differ between the parental lines. We demonstrate that these deviations are affected by genetic variants in interaction with the environment, and hypothesize that geographic differences in gbM arise through the inclusion of such deviations in the zygote.

## Introduction

DNA (cytosine) methylation is an epigenetic mark associated with transcriptional regulation, in particular transposable-element silencing. Unlike animals, where methylation is mostly found on CG sites, cytosines sites in plants are also methylated in other contexts: CHG and CHH, where H = A, C or T. Non-CG methylation is mainly present on transposable elements and is associated with repression of transcription. It cannot be directly inherited, is found on only a fraction of cells, responds to the environment, and has been shown to be influenced by trans-acting genetic loci in *A. thaliana* [1–5]. This is in sharp contrast to CG methylation (mCG), which is maintained during DNA replication through the action of MET1, the homolog of mammalian DNMT1. Unlike in animals, mCG is not reset every generation in plants, but shows stable transgenerational inheritance [6–10]. As in animals, mCG in plants is present not only on transposable elements and other heterochromatic regions, but also on the coding regions of a subset of genes, a phenomenon known as gene-body methylation (gbM) [11–14]. Genes with gbM tend to be evolutionarily conserved and constitutively expressed, *i.e.*, they are "house-keeping genes". Although it has been argued that gbM is under selection [15, 16], its function is unclear [17–20].

What is clear is that mCG levels vary greatly between natural inbred lines of *A. thaliana*, and that the pattern of variation reflects the geographic origin of the lines and is correlated with various climate variables [3, 21, 22]. In particular, plants that originate from the colder climate of northern Sweden have higher gbM levels than plants from warmer regions [3]. There are several possible explanations for these patterns.

The first is that gbM is under direct selection [3, 16]. The weakness of this explanation is the lack of evidence for any mechanism whereby selection could affect gbM at thousands of loci across the genome [20].

Alternatively, plants could simply retain an epigenetic memory of their ancestral climate. However, for this to work, the environment has to affect DNA methylation. Numerous studies have examined the effect of growth conditions on DNA methylation by growing plants in different environments, and while there is clear evidence that non-CG methylation responds strongly to the environment, mCG seems quite stable, at least at the genome-wide level [3, 23–26], consistent with its apparent stability over large numbers of generations [8, 9, 27, 28].

Finally, the geographic pattern of gbM could be due to trans-acting genetic variation. Indeed, genome-wide association studies (GWAS) have identified several loci affecting non-

CG methylation [3–5, 21, 29], and it possible that mCG could have been similarly affected by trans-acting modifiers. However, because mCG is stably inherited, it is not a phenotype, and the present methylation state of an individual reflects not only its current genotype but also the history of its genome. As a consequence, unless the genetic effects are very strong, most of the variation will reflect (a complex and highly non-random) history rather than genetics, making mapping of genetic modifiers difficult. It is therefore not surprising that GWAS found no evidence for genetic variants influencing mCG [21].

This paper looks for evidence of genetic variants influencing gbM using an F2 cross between a northern Swedish line with high gbM and southern Swedish line with low gbM. To also look for environmental effects, the experiment was carried out at two different temperatures, 4˚C and 16˚C, and the cross was reciprocal to investigate possible parent-of-origin effects, which are *a priori* plausible [30–32]. After excluding about 3 k genes with evidence of non-CG methylation (characteristic of transposable elements), we consider all exonic CG sites (a total of over 650 k). Our hope was that our large sample size (a total of over 600 F2 individuals were bisulfite-sequenced) might allow us to detect minute changes in mCG despite its stable inheritance.

## Results

### Residual heterozygosity in one parental line

The bisulfite-sequencing data were used to genotype the F2 populations. While doing so, we discovered that one of the parental lines had harbored residual heterozygosity: there are at least two Mb-length regions segregating between the putatively reciprocal F2 populations (S1 Fig). This is irrelevant within each cross, because a single F1 parent was used to generate each F2 population, however, it makes interpretation of differences between the two cross-directions challenging, because they could be due to parent-of-origin effects or genetic differences. For this reason, we will initially focus on the cross in which the northern line was used as mother while the southern was used as father (n = 308; S2 Fig), and discuss the (partially) reciprocal cross later. When analyzing parental lines, which were grown in replicate (S1 Table), the segregating regions were eliminated from the analysis.

### Differences in gbM between the parental lines

Methylation estimates from bisulfite sequencing are noisy for a variety of experimental reasons, the most obvious one being low sequence coverage of a possibly heterogeneous population of cells. However, the parental lines were grown in replicate in both temperatures, allowing us to estimate the grandparental state, confirm that gbM is highly consistent between replicates, and that individual sites are either methylated (nearly 100% methylation across sequencing reads) or unmethylated (approximately 0% methylation across sequencing reads), consistent with direct inheritance through both mitosis and meiosis, leading to a cell population with minor deviations from the inherited state, largely independent of temperature (Fig 1 and S3 Fig).

The analysis also demonstrated that the previously reported difference in average gbM level between these lines [3] is mostly due to more sites being methylated in the north (rather than a quantitative difference across many sites). Of the roughly 25% of sites that are methylated in at least one of the parental lines, approximately 45% differ between the parental lines, and, of these, 70% are only methylated in the northern line (Fig 1).

### Inheritance of gene-body methylation in the F2 population

In the F2 population we do not have replication of entire genotypes, but we have roughly 75-fold replication of the genotype at each site, because 1/4 of the 308 F2 individuals are

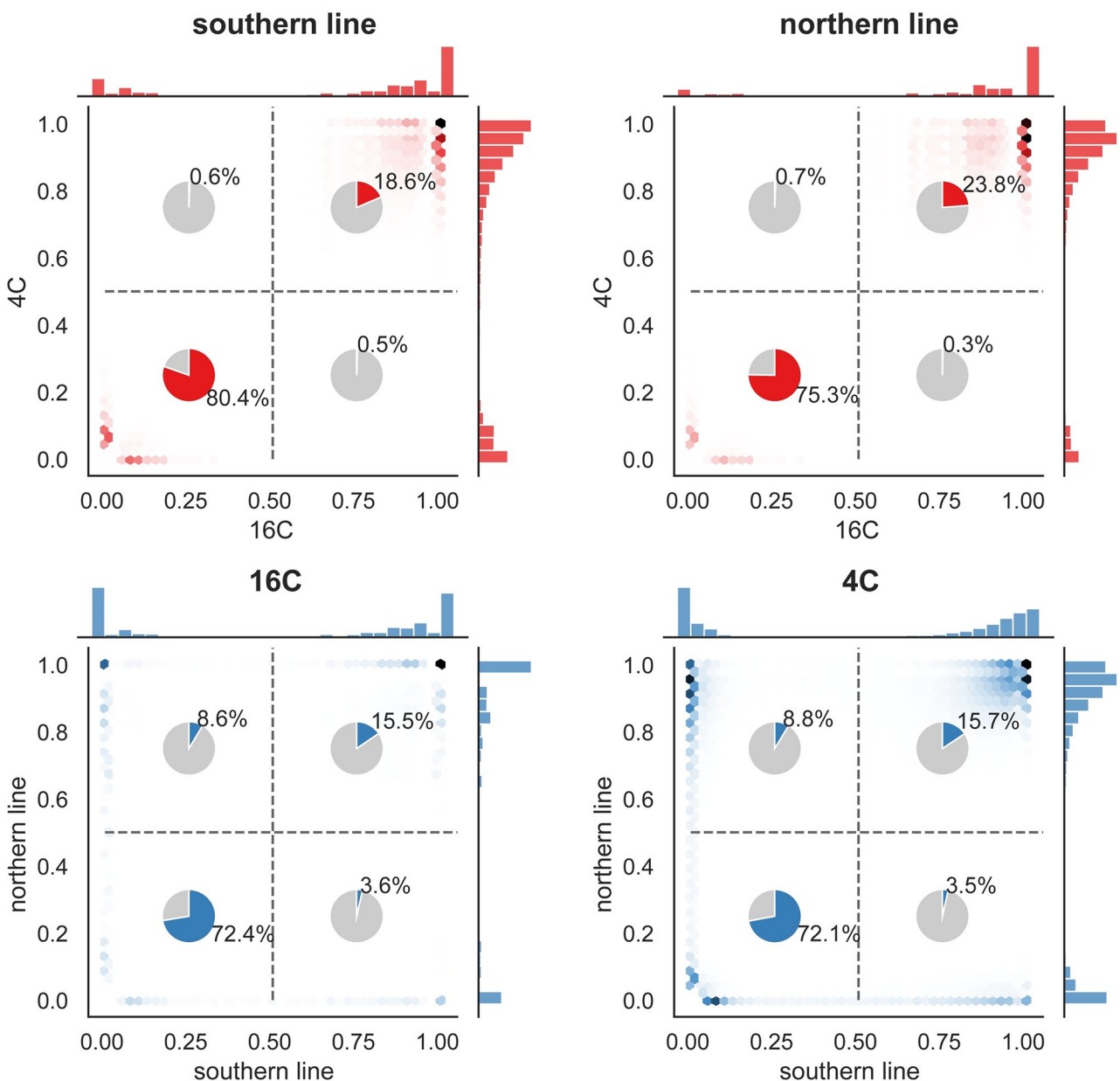

**Fig 1. The pattern of gbM across sites.** The plots show the distribution of average methylation levels across 650,595 potential gbM sites at 4˚C and 16˚C, separately for the two parental lines. The pie charts shows the fraction of sites classified as methylated or unmethylated using 50% methylation as a cut-off (see Materials and methods for details). The top plots compare temperatures for each parental line; the bottom plots compare parental lines for each temperature.

expected to be homozygous for northern ancestry (NN) at each site, 1/4 to be homozygous for southern ancestry (SS), and 1/2 to be heterozygous (NS). Ancestry can accurately be inferred using SNP haplotypes, and by combining this with the methylation states in the F2 population we can also infer the epigenotype at each site in the F1 parent—and confirm that gbM shows the expected Mendelian segregation (S3 Fig, [33]).

The inferred F1 epigenotype can be compared with the inferred grand-parental epigenotype to get an estimate of the epimutation rate. This is not straightforward and requires a number

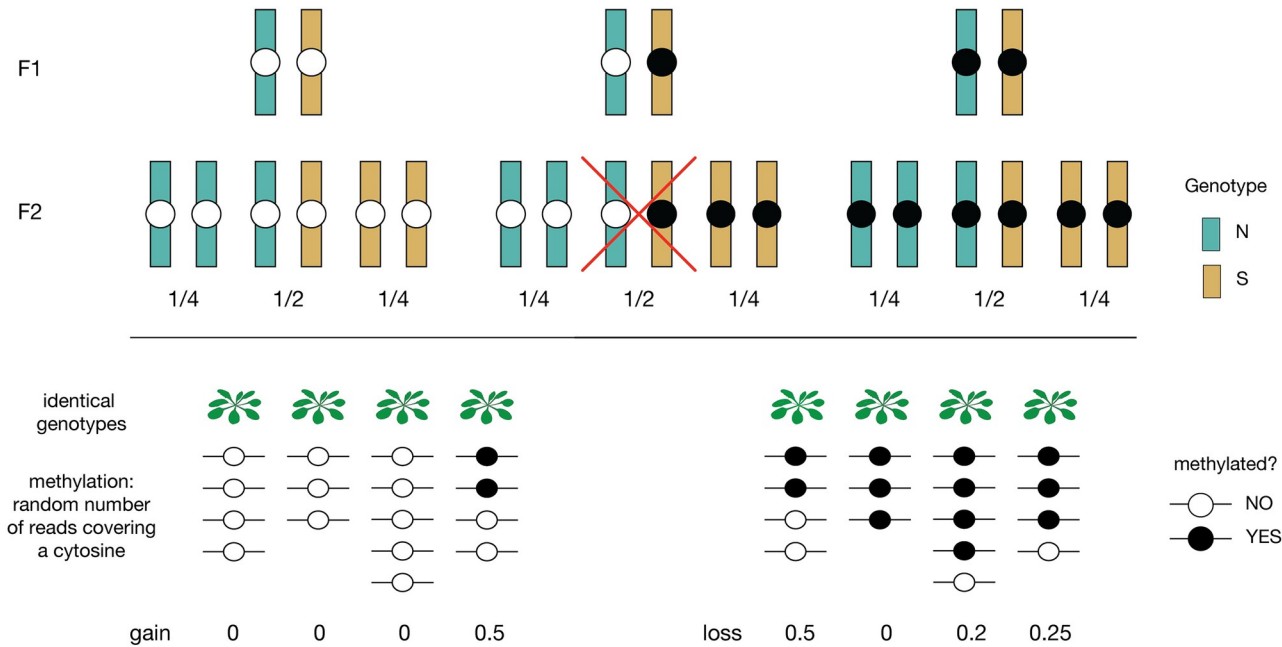

**Fig 2. Quantifying somatic gains and losses.** In the F2 population, each gbM site is characterized by ancestry: NN, NS, and SS. Independently of this, there are three types of sites: those for which F1 parent was homozygous unmethylated, those for which it was heterozygous methylated/unmethylated (could be either on N or S allele), and those for which it was was homozygous methylated. In the F2 population we estimate gains only for individuals that should have inherited the homozygous unmethylated state, and losses only for individuals that should have inherited the homozygous methylated state. We do not use individuals heterozygous for methylation. Different analyses then use different subsets of the gain/loss data as detailed below.

of assumptions because differences could have arisen at any point across two generations—and could also reflect heterozygosity in the grand-parental generation, as well as various artefacts that are difficult to control for. We obtain a per-generation, per-site rate of loss of gbM of ∼ 0.2%, and corresponding rate of gain of ∼ 0.04%, although we caution that there are aspects of our data we cannot explain (see Materials and methods for details). We will return to these in the Discussion.

However, these epimutations did not occur in the F2 generation. While they may have been affected by the F1 genotype, they do not reflect genetic variants segregating in the F2 population, nor our temperature treatment. In order to take advantage of the experimental design, we need to focus on changes that happened in the F2 generation itself, *i.e.*, we need a proper phenotype. Thus we focus on somatic deviations from the inherited state (as seen in the parents in Fig 1). These are by definition phenotypes affected by genotype and environment, and while the deviation at a particular site in a particular individual is very poorly estimated (primarily due to insufficient sequencing coverage), this is compensated by the size of the F2 population. It is obvious from Fig 1 that *gains* (positive deviations from an inherited state of 0% methylation) have a very different distribution from *losses* (negative deviations from an inherited state of 100% methylation), and we therefore estimate each separately, as explained in Fig 2.

## Losses and gains reflect different processes

Somatic losses and gains differ in several aspects. First, estimated losses are on average two orders of magnitude higher than estimated gains: 7.3% vs 0.09%, respectively (S2 Table). Second, gains and losses show very different distributions across the genome (gains are 2.8 times

higher in pericentromeric regions), similar to what has been observed for transgenerational epimutations [34, 35], and are also associated with different chromatin states (S4 Fig). Third, while both gains and losses are (weakly) negatively correlated with gene expression, this correlation is much more pronounced for loss (S4 Fig). Fourth, losses vary much more across the four possible CG contexts (CGA, CGT, CGC, CGG) than gains. In particular, losses are 22% higher on CGT compared to the other contexts (S5 Fig).

Gains and losses are only weakly affected by temperature (Figs 3 and S5). They do, however, depend on local ancestry: on average losses are 2% higher on SS alleles compared to NN alleles, while gains are higher 29% on NN alleles than SS alleles (although the pattern varies greatly across the genome; see Fig 3). Potential causes for these patterns will be discussed below. Finally, both gains and losses exhibited positive auto-correlation along the genome (gains are correlated with gains at nearby sites, and the same for losses). We do not observe any such effects on non-CG methylation (S5 Fig).

Importantly, both gains and losses are higher for sites that differ between the two parental lines: the increase is roughly 10-fold for gains and almost 2-fold for losses (Fig 3C and S3 Table). Given this, and the other similarities to transgenerational epimutations noted above, it is reasonable to speculate that both reflect the same mechanism, and that transgenerational epimutations are simply a subset somatic epimutations that end up being transmitted via gametes.

Motivated by this, we investigated whether the observed gains and losses have the properties one would naïvely expect of mitotically heritable epimutations. Specifically, we tested whether cells switch state independently of each other (conditional on estimated rates of switching) within and between individuals using a simulation approach (see Materials and methods). If deviations were largely due to somatically inherited epimutations (perhaps affecting large sectors of the sequenced plants), changes within plants would be positively correlated, and we might see inflated variance between plants, with some plants being responsible for most of the average deviation at a given site (see Fig 2). However, with the possible exception of gains on sites that differ between the parents, we see no evidence of this phenomenon (S6 Fig). The distribution of gains seems compatible with independent changes within and between plants, and there is no evidence for large sectors due to somatic inheritance (*n.b.* our power to detect such sectors is extremely limited due to low sequencing coverage per-individual).

The distribution of losses, on the other hand, is clearly incompatible with independent mutations, but in the opposite direction: there is far too little variation between individuals for losses to reflect random independent events (S6 Fig).

### Genetic architecture of deviations

To investigate genetic and environmental factors influencing these deviations, we used a standard F2 linkage mapping model that includes temperature as an environmental factor and allows for genotype-by-environment interaction (G×E). As phenotypes, we used deviations in 500 kb windows across the genome. Windows were used because per-site deviations are far too noisy (since deviations are rare), and using genome-wide deviations is inappropriate given clear evidence for heterogeneity across the genome (Fig 3). The results provide further evidence that gains and losses are different phenomena. For both phenotypes, we identify significant trans-acting QTL, but they appear to be different (Fig 4A and S4 Table; note that while the confidence intervals on chromosome 5 overlap, the peaks are far apart and affect different loci). Furthermore, gains are also affected by strong cis-acting factors.

QTL affecting losses are far stronger and had more consistent effects across the genome. We identify two major QTL accounting for about 5% of the variation each, with similar effects

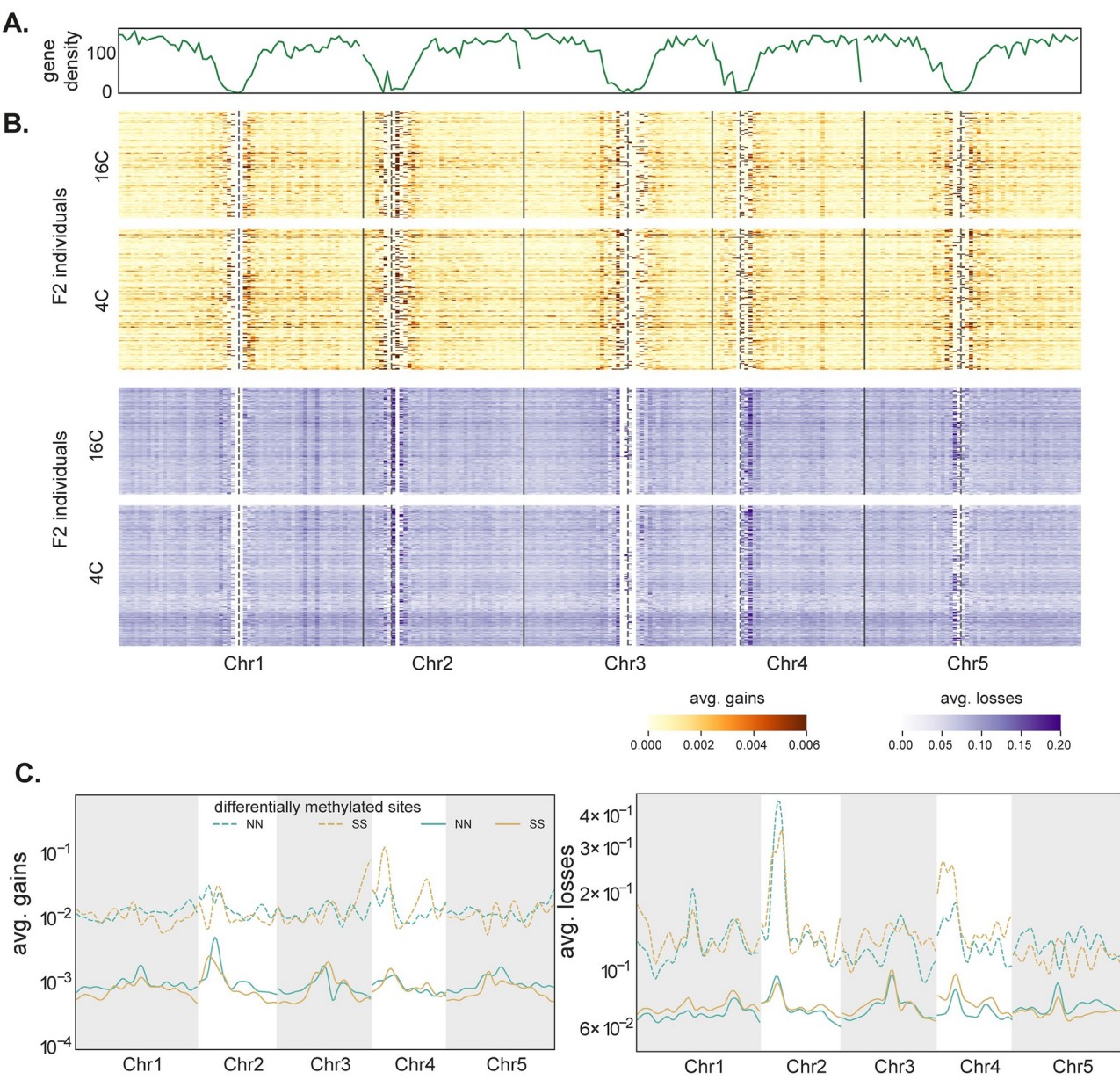

**Fig 3. Somatic deviations across the genome. (A)** Line plot for gene density in 500 Kb windows. **(B)** Heatmaps show genome-wide somatic deviations for gains and losses for genes in F2 individuals at both temperatures (n = 308). Each row is an individual. Gene density and deviations were calculated in 500 kb windows across the genome. Vertical solid lines represent chromosome breaks and dotted lines represent the centromere positions. **(C)** Average gains and losses across the genome for homozygous NN and SS individuals. Deviations at sites that differ between the parents are shown using dashed lines (see Fig 2).

in both temperatures, and with additive effects within and between loci (*i.e.*, no dominance or epistasis; see Fig 5, S7 Fig and S4 Table). The northern and southern alleles have opposite effects at the two loci.

The two QTL for gains affect different windows (Fig 5). Each QTL explains a couple of percent of the variation, and the north-south direction of effects is again reversed between the loci. At each locus, the allele associated with greater gains is recessive, and the effect of the chromosome 5 QTL is only seen at 4°C. There is no evidence for epistasis.

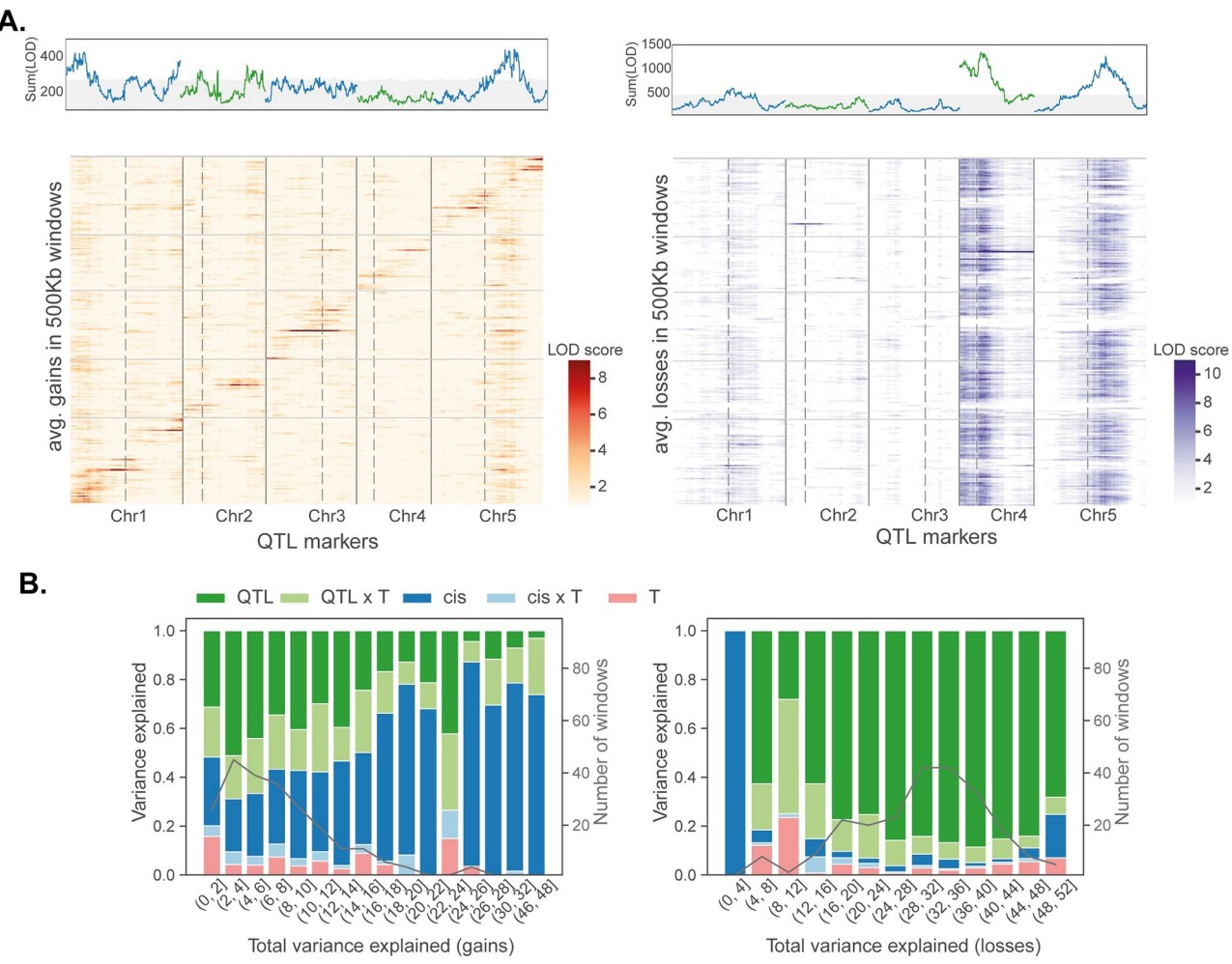

**Fig 4. Genetic architecture of deviations. (A)** Heatmaps showing linkage mapping results for gains and losses in 500 kb windows together with plots summing LOD scores across these windows. Peaks above gray region are significant using a 1% FDR based on genome permutations. Vertical dotted lines identify centromeres and solid lines separate chromosomes. **(B)** Bar plots summarizing variance partitioning results for gains and losses. Results are binned by total variance explained, with thin black lines showing the distribution of windows across bins.

In order to quantify the factors affecting the deviations, we partitioned the variance in each 500 kb window using a linear model that includes local (cis-) genotype (*i.e.*, NN, SS, or NS), temperature, and the identified QTL as factors. The results for gains and losses are again strikingly different (Fig 4B). For losses, the majority of the variance explained is by the QTL, with a minor role for QTL-by-temperature (QTL×T) interactions. For gains, QTL, QTL×T, and cis-genotype appear to play roughly equal roles, and there is also evidence for interactions between the cis-genotype and temperature. Temperature, in-and-of-itself, explains little of the variation, however, the G×E effects for gains are substantial. This can also be seen in the predicted response for the parental QTL genotypes (Fig 5D), which agree with direct estimates (S8 Fig).

In an attempt to fine-map some of the QTL identified here, we turned to GWAS. We used the population data from reference [3], where about 100 accessions were grown at two temperatures, 10˚C and 16˚C. We calculated genome-wide deviations for each accessions by considering sites with less than 50% methylation as gains and sites with more than 50% methylation

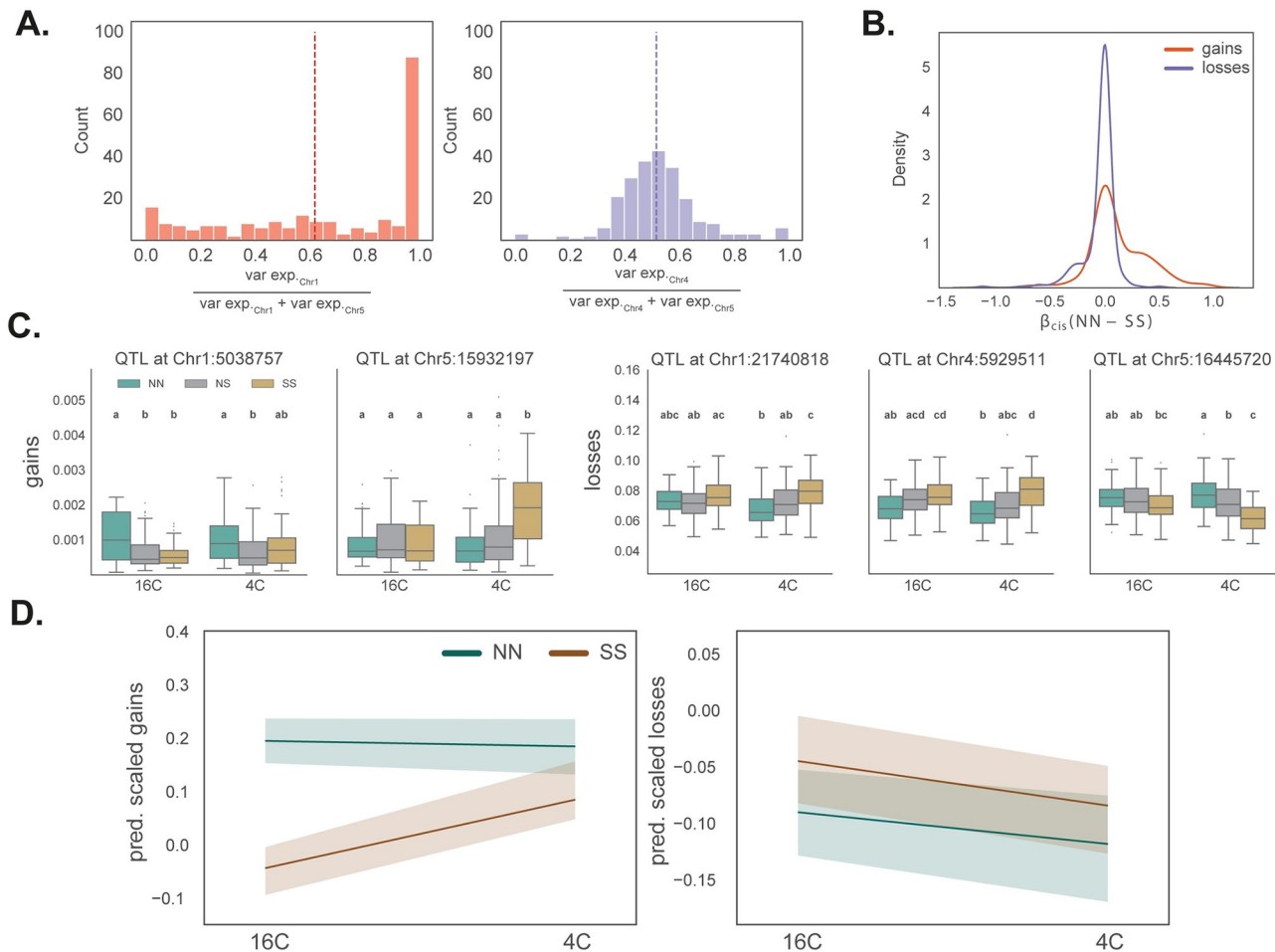

**Fig 5. QTL effect-size estimates.** **(A)** The distribution of variance explained across 500 kb windows for a gain QTL (left) and a loss QTL (right). The mean effects (vertical lines) are similar, but the gain QTL has a highly skewed distribution, with strong effect only on a subset of windows, where the loss QTL affects most of the genome. **(B)** The distribution of cis effect sizes. **(C)** Genotypic effects for two gain QTL and three loss QTL. Average gains and losses across windows significantly associated with QTL are shown. **(D)** Reaction norms for predicted gains and losses in individuals homozygous for the northern or southern alleles at all significant QTLs.

as losses. Consistent with temperature having little effect, deviations are highly correlated between the two temperatures ($r = 0.74$, S10 Fig). The average gains and losses across accessions are around 0.5% and 9%, respectively, and we performed GWAS using these as phenotypes, but could not identify any significant associations (S10 Fig). The same is true when 500 kb windows rather than genome-wide averages are used.

## Cis-effects on deviations

We have seen that deviations are associated with the local haplotype, *i.e.*, they are affected by cis-acting factors (Fig 4). The effect is particularly pronounced for gains, but is also seen for losses. Generally speaking, the cis-effects work in the direction of the observed differences, *i.e.*, gains are more pronounced on the more methylated N allele and losses are higher on the less methylated S allele (Fig 5B).

While it possible that these effects are due to genetics, it would imply that cis-regulatory differences have evolved throughout the genome. It seems more likely that the effects are a

consequence of the epigenetic differences that we know exist. As mentioned previously, deviations are associated with the underlying chromatin state (S4 Fig), suggesting the local epigenetic state influence them.

Motivated by this, we examined whether deviations are correlated with methylation levels at the level of individual genes. And indeed, gains tended to be higher on the allele with higher methylation level, while losses show the opposite pattern (Fig 6A). Zooming in further, we find that both gains and losses are strongly affected by nearby methylation at a nucleotide scale (Fig 6B). For gains in particular, the effect seems to be limited to less than 30 bp. (Note that these findings demonstrate non-independence between nearby sites along the chromosome, which is very different from the non-independence between plants discussed above in relation to S6 Fig.)

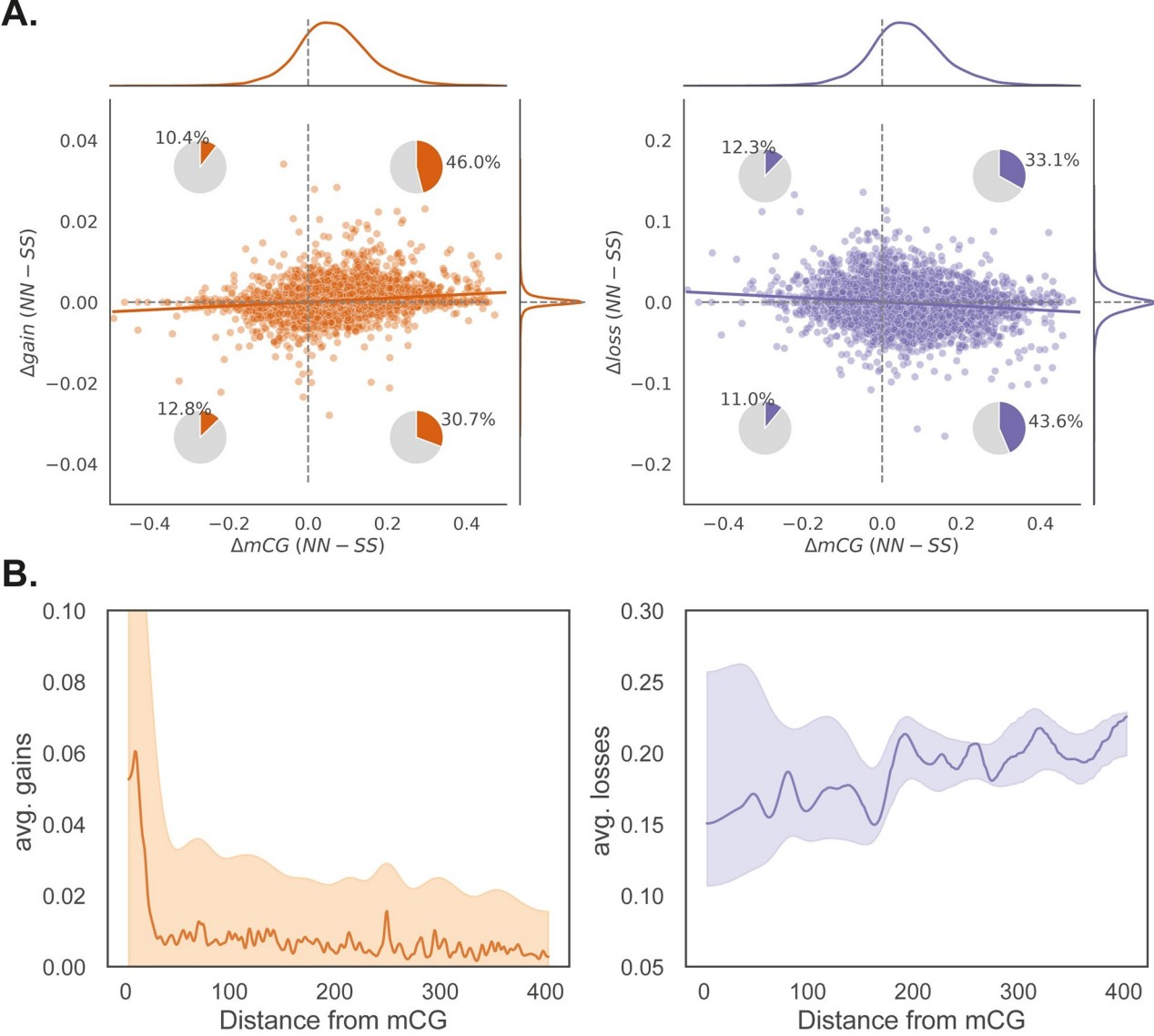

**Fig 6. Cis-effects on deviations. (A)** Across genes, the difference in gains between northern and southern alleles (estimated from homozygous individuals) is correlated with the difference in gbM between the same alleles. Both correlations (Spearman coefficients of 0.2 and -0.12 for gains and losses respectively) are significant ($p < 0.01$). **(B)** Average gains and losses at a given CG site depends on the distance to the nearest methylated CG site. The lines represent the median across the mCG sites and the colored region show the range of the central 90% of the data. Data is shown only for sites in an example region of 2 Mb on Chromosome 1.

## Partially reciprocal cross

As noted above, this experiment was designed to include a reciprocal cross in order to test for parent-of-origin effects on methylation, but undetected residual heterozygosity in one of the parental lines made the cross only partially reciprocal, making interpretation of differences challenging. For this reason, discussion thus far has been limited to the cross in which the northern line was used as maternal parent.

In the reciprocal cross, we observe very similar patterns of deviations across the genome (S9 Fig). Average losses are strongly correlated between the F2 populations at the level of genes (Fig 7A), and the two significant QTL appear to be replicated (although the significance of the one on chromosome 5 was weaker). In addition, we identify a new QTL on chromosome 1 that directly overlapped the region segregating between the F2 populations and is thus probably due to a genetic difference rather than the direction of the cross (Fig 7B).

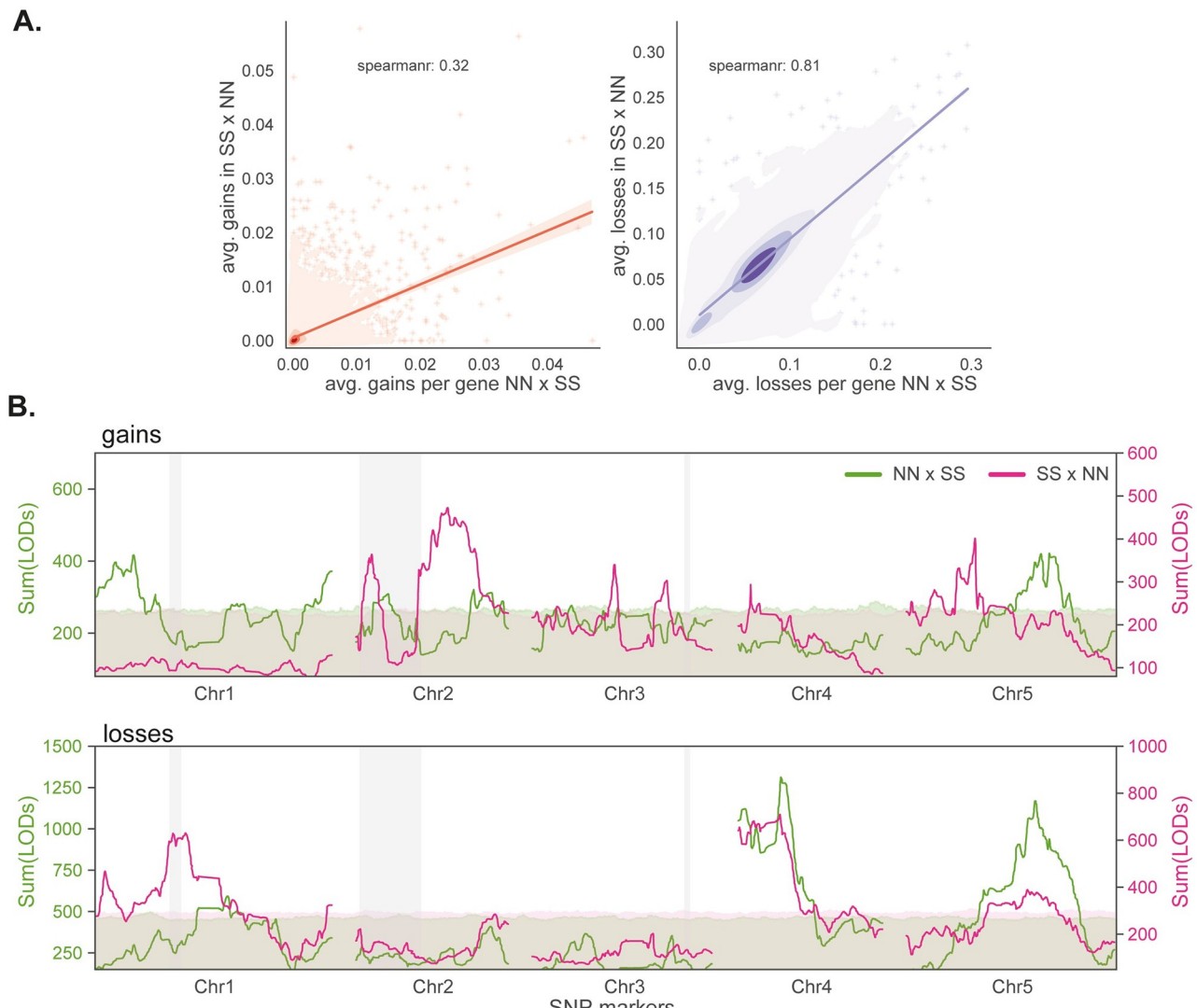

**Fig 7. Somatic deviations in the partially reciprocal cross. (A)** Average gains and losses per gene (in NN background) for both directions of the cross. **(B)** Aggregate linkage mapping results (sum of LOD scores for 500 kb window) for both directions of the cross. The results for the NN×SS direction were already shown in Fig 4A. The 99% significance thresholds were determined using 1000 genome rotations (see Materials and methods). Regions that segregate between the two F2 populations are shown using grey vertical bands.

In stark contrast, average gains are not strongly correlated between the two directions of the cross (Fig 7A). Given this, it is not surprising that the corresponding QTL mapping results are also discordant, with previously identified QTL being replaced by different ones in (Fig 7B). The QTL do not overlap regions that segregate between the F2 populations, and must thus either reflect epistatic interaction with putative causal polymorphism in these regions, or parent-of-origin effects.

## Discussion

### Somatic gains and losses

The goal of this study was to gain insight into how gbM is inherited—and how it changes. In particular, we sought evidence that trans-acting genes and environmental factors could influence gbM, which might help explain observations like the large differences between northern and southern Sweden [3].

We decided to use a traditional diallel F2 cross between a northern and a southern line, grown at two different temperatures. The obvious problem is that gbM is not really a phenotype—in the sense that the methylation state at a particular site in a particular individual was simply inherited, and hence does not directly reflect the genotype or environment of that individual. Indeed, our study provides very strong confirmation that gbM shows the expected Mendelian segregation [33]. To have a phenotype that we could map, we instead focused on *deviations* from the inherited methylation state; either *gains* (for sites inherited as unmethylated) or *losses* (for sites inherited as methylated). To the extent these reflect somatic deviations in the F2 generation (a question to which we shall return below), they would be proper phenotypes, reflecting the genotype and environment of the individual in which they were found.

Gains and losses behaved very differently. The former occur at low rates (higher than the estimated transgenerational rate of gains [28], but same order of magnitude: 0.09% vs 0.025%), independently within and between individuals; perfectly consistent with their being somatic epimutations. Like transgenerational gains of gbM, they were also clustered and found at a higher rate in pericentromeric regions [34], as well as positively correlated with nearby (within 30 bp) methylation (*cf*. Fig 6B with Fig 3 in reference [36]). It is thus reasonable to hypothesize that they are somatic epimutations, and reflect the same mechanisms that give rise to transgenerational epimutations. If this is true, the QTL we identified correspond to *bona fide* modifiers of the epimutation rate—which makes it very interesting that they show strong G×E effects (Fig 4), as well as parent-of-origin effects (Fig 7).

Losses, on the other hand, look nothing like somatic epimutations, at least not at first. They occur at rates two orders-of-magnitude higher than transgenerational losses (7.3% vs 0.063% [28]), and are furthermore positively correlated between individuals, which is clearly not consistent with random mutations. One explanation is that they reflect experimental artefacts due to bisulfite sequencing, which typically shows less than 100% methylation when used on fully methylated control DNA [37]. However, while this is likely to contribute, artefacts would not give rise to highly significant QTL. The losses must also have a biological basis, but not necessarily one related to epimutations. For example, mCG is automatically lost during DNA replication (the newly synthesized strand is unmethylated, leading to hemi-methylated DNA), and the maintenance of mCG across mitosis is therefore an active process, dependent on MET1 [1]. Anything that caused an imbalance between cell division and MET1 activity could lead to apparent somatic losses, and these could well be unrelated to transgenerational epimutations.

This said, there is also evidence that the losses we observed are related to transgenerational epimutations. Gains and losses both are much more common on sites that differ between the

parental lines (Fig 3), and they also show a dependence on local methylation that is similar to what has been seen for transgenerational epimutations (Fig 6B).

## Sex and epimutations

We were also able to obtain rough estimates of the transgenerational epimutation rate by comparing the F1 epigenotypes (inferred from the F2 generation) with the grand-parental epigenotype (inferred from replicate parental individuals). These estimates were complicated by a number of technical problems (see Materials and methods), and we emphasize the study was not designed for this purpose. Nonetheless, the results are interesting, and we discuss them here because we think they may help interpret our main findings.

The average rate of transgenerational gains was 0.04%, which is broadly consistent with published estimates of transgenerational gains as well as with our somatic gains. The average rate of transgenerational losses was 0.2%, which is far more consistent with published estimates, albeit still high, than our somatic loss-rates, which 30 times higher still—a further demonstration that the latter must include phenomena other than somatic epimutations. However, far more interesting is the several lines of evidence that epimutation is not a random process (S14 and S15 Figs, and S5 Table): first, we observed very strong direction-of-cross effects; second, although the pedigree encompasses two generations, a disproportionate fraction of mutations appear to have occurred in the F1 generation, and; third, there was an order of magnitude more overlap in epimutated sites between the two crosses than could be expected under any model of random mutation.

Although initially perplexing, these observations are reminiscent of several studies of non-additive methylation in *A. thaliana* "hybrids" [38–40], which reported massive (and at least partly reproducible) changes in DNA methylation in F1 plants from crosses with different inbred lines, and demonstrated that they were caused by trans-acting factors. Although most of the changes affected transposable elements and were shown to be dependent on the RdDM pathway and trans-acting 24-nt siRNAs, not all were. Furthermore, at least some of these changes were shown to persist in the F2 generation. A more recent study has also demonstrated genome-wide "methylome remodeling" in F1 plants and attributed it to trans-acting factors [41].

These studies remind us that there is every reason to expect epigenetic changes in relation to sex. While the phenomenon has often been framed in terms of some kind of temporary "hybrid vigor" [42], it is important to remember that sex is actually normal, even in selfers, and that every outcrossing event will lead to a unique genome with an epigenetic profile that may not be in equilibrium with its complement of trans-acting modifier alleles (whatever these might be). We think it is plausible that the relatively high epimutation rates we see, somatically as well as trans-generationally, reflect the fact that we are are looking at outcrossed individuals, whereas published estimates come from inbred lines, which may be closer to equilibrium (to the extent such a thing exists).

## Conclusions

The rationale for this study was to try to understand what could have caused genome-wide geographic differences in gbM such as those seen in Swedish *A. thaliana* [3]. If we believe that mCG evolves in a clock-like manner analogous to DNA sequence evolution [8–10, 20, 43], this observation requires genome-wide selection on gbM at individual loci, which would be implausible even if we had a clear selective mechanism, which we do not [3, 16, 20].

Here we suggest an alternative explanation, namely that mCG is affected by trans-acting genetic modifiers in interaction with the environment, and that the methylation state of a

particular genome at a particular point in time is an epigenetic memory of past genomes and environments. We further propose that epimutation rates estimated from mutation accumulation (MA) lines may be misleading because many epigenetic changes are driven by genetic and environmental perturbations, and it is far from clear whether (and in what sense) they should be viewed as random [35, 36, 39, 40, 44]. Indeed, the term "epimutation" may in many ways be a misnomer.

Note that this second explanation is agnostic about the adaptive importance of mCG, but cautions against considering it simply a "fifth base" (with regulatory function), that can be understood using the standard modeling framework of molecular population genetics.

To test this hypothesis we need multi-generational pedigree studies of outcrossed individuals, ideally with sufficient sample size to map any causal loci. A major challenge, common to all studies of mutational processes, is that changes may be very rare. Even the relatively high epimutation rates estimated here are in the $10^{-4}$–$10^{-3}$ range, implying that the observed difference in gbM between northern and southern Swedish lines would take at least a thousand years to accumulate—which is plausible given the migration history of *A. thaliana* [45]. However, slow change could help explain the lack of divergence between North American *A. thaliana*, which were introduced by Europeans fairly recently [27]. If these difficulties could be overcome, major conceptual advances in our understanding of methylation might be made.

## Materials and methods

### Plant growth

We chose two natural inbred lines from Sweden that had been shown to differ considerable in gbM [3]: one line from Lövvik in northern Sweden (ID 6046, lat. 62.800323, long. 18.075722) with average gbM of 16% and another from Drakamöllan in southern Sweden (ID 6191, lat. 55.758856, long. 14.132822) with average gbM of 12.5%. We generated recombinant hybrid progeny of these two lines by collecting seeds from selfed F1 individuals for the reciprocal directions (S1 Fig). Selfed parental lines were grown along with F2 individuals from two families at two temperatures (16°C and 4°C) in a randomized block design (S1 Table). We grew plants on soil and stratified for 5 days at 4°C in the dark before transferring them to long day chambers with 16 hours of light and 8 hours of darkness. When plants attained the 9-true-leaf stage of development, one or two leaves were collected and flash-frozen in liquid nitrogen.

### DNA extraction and bisulfite sequencing

Genomic DNA was extracted from frozen tissue using the NuclearMag Plant kit (Machery-Nagel). We adopted a tagmentation-based protocol to generate multiplexed DNA libraries for whole-genome bisulfite sequencing (T-WGBS; [37]). We optimized the protocol for low DNA inputs (20 ng) and high-throughput (96-well plates). We used in-house Tn5 transposase generated at Vienna BioCenter Core Facilities. The tagmentation, oligonucleotide replacement and gap repair were done according to the T-WGBS protocol.

We used EZ-96 DNA Methylation-Gold Mag Prep kit (Zymo Research) for bisulfite conversion from tagmented DNA. We PCR-amplified bisulfite-treated DNA with 15–16 cycles with KAPA HiFi Uracil polymerase (Kapa Biosystems). We used Illumina TruSeq unique index adapters for PCR amplification and multiplexing of the libraries. Amplified libraries were validated using Fragment Analyzer Automated CE System (Advanced Analytical) and multiplexed (96X) in equimolar concentration. Libraries were sequenced on Illumina HiSeq 2000 Analyzers or HiSeqV4 using the manufacturer's standard cluster generation and sequencing protocols in 100–125bp paired-end mode.

## Sequencing data analysis

Sequenced BS-seq reads were analyzed using a well-documented nf-core pipeline (github.com/nf-core/methylseq). First, BS-seq reads were trimmed for adaptors using cutadapt (default parameters), and we clipped 15 bp at the beginning of the reads due to uneven base composition. Second, the trimmed reads were mapped to the TAIR10 (Col-0) reference assembly using bismark relaxing mismatches to 0.5 [46]. Third, methylation calling was performed using methylpy on the aligned bam files. We used custom scripts to calculate weighted averages of methylation [47] at annotated genes and transposable elements using the ARAPORT11 annotation (www.arabidopsis.org/download/index-auto.jsp?dir=%2Fdownload_files%2FGenes%2FAraport11_genome_release). All scripts used were packaged in python and are available on github (github.com/Gregor-Mendel-Institute/pyBsHap.git).

## Bisulfite conversion rate estimation

It is common practice to use the chloroplast genome (cpDNA) to estimate conversion rates for BS-seq libraries, since cpDNA is unmethylated [48]. The non-conversion rate was calculated as the fraction of methylated cytosines from reads mapped to the cpDNA. The estimated conversion rate for the libraries is on average 99.7% (S11 Fig).

Incomplete bisulfite conversion would result in spuriously methylated sites across the genome, mimicking somatic gains of methylation. Hence we ignored methylation on sites that did not have significantly higher methylation than expected due to non-conversion (using a binomial test with probability of 0.3%; p-value of 0.05). Although this reduces the sensitivity in estimating somatic deviations, it increases the accuracy. Similarly, somatic losses might be affected by over-conversion due to bisulfite treatment, but here we have no good estimate of the rate.

## Gene body methylation

We calculated methylation levels on all exonic CG sites. We excluded genes with significant non-CG methylation in either of the parental lines from the analysis (S12 Fig), but did not rely on any other epigenetic marks. In doing so, the average mCHG and mCHH levels per gene were scaled, and outlier genes were identified using twice the standard deviation. This resulted in a total of 24,841 genes from the original 27,445 annotated in Araport11. The gene number was reduced by another 100 when we consider non-CG methylation on the introns. We used mCG sites in these filtered gene set for the entire analysis.

We did not exclude genes based on evidence of non-CG methylation in the F2 generation, since we explictly wanted to test the hypothesis [18] that changes in mCG correlate with changes in non-CG (especially CHG). No such effect was found (S5 Fig).

## SNP calling and genetic map reconstruction using bisulfite treated libraries

The mapped bam files from bismark were modified for base positions that are influenced by bisulfite treatment ($C \rightarrow T$ and $G \rightarrow A$) using Revelio (github.com/bio15anu/revelio.git) [49]. We genotyped 10.7 million previously identified SNP sites [50] using bcftools with default parameters [51]. The scripts for the pipeline were packaged and hosted on github (github.com/Gregor-Mendel-Institute/nf-haplocaller).

Next, we inferred the underlying ancestry at each SNP marker segregating between parents in F2 individuals using a multinomial hidden Markov model (adapted from reference [52]) packaged in the SNPmatch package (github.com/Gregor-Mendel-Institute/SNPmatch.git). Bisulfite sequencing gives uneven coverage across the genome, but such data

can be used to infer ancestry with high accuracy, in particular in F2 individuals were ancestry tracts are very long. We filtered out SNP markers having identical genotype data across individuals using R/qtl package [53]. This resulted in a total of 3983 SNP markers used for linkage mapping.

### Residual heterozygosity in reciprocal cross

We calculated percentage of heterozygous SNP calls for parental lines sequenced as part of the 1001 Genomes project [50]. At least four genomic regions more than 300 kb had residual heterozygosity in the southern parent (S1 Fig).

As a consequence, for any given site in these regions, different southern alleles could be segregating in the reciprocal crosses, *i.e.*, rather than $N$ and $S$ alleles segregating in both, we could have $N$ and $S_1$ in one direction, and $N$ and $S_2$ in the other. To identify such regions, we identified all SNP segregating in each F2 population, then compared them using SNPmatch [54]. As expected, this revealed that a subset of the putatively heterozygous regions differed between the directions of the cross (S1 Fig).

### Estimating somatic deviations

Each F2 family (NN×SS and SS×NN) is the offspring of a single F1 individual, a hybrid with NS-ancestry at every site. Every mCG site would either be methylated (11), unmethylated (00) or heterozygous (01) in this F1 individual (Fig 2). Due to the stable inheritance of mCG, we expect the parental methylation state to have been passed on, and this was readily confirmed. Somatic gains and losses were calculated as weighted averages across sites classified as having been inherited homozygous unmethylated or methylated, respectively [47]. This was either done per gene or in windows of 500 kb. We chose 500 kb as it resulted in averaging gains and losses across $\sim 10^4$ cytosines, which is appropriate given epimutation rates in the order of $10^{-4}$. We also averaged the deviations across ten annotated gene clusters based on chromatin state [55].

In individuals heterozygous for methylation state (NS), we expect to see 50% methylation since we lack the power to do allele-specific methylation (given 100 bp reads, and our data supports this (S3 Fig).

The python scripts used for these analyses were packaged and are hosted on github (github. com/Gregor-Mendel-Institute/pyBsHap).

### Modeling somatic deviations between individuals

Let $s_{ij}$ be the number of reads with ancestral methylation at site $i$ in individual $j$, and let $f_{ij}$ be the number of reads with non-ancestral methylation. We calculate deviation from the ancestral state as $x_{ij} = f_{ij}/n_{ij}$, where $n_{ij} = s_{ij} + f_{ij}$. We also define the average deviation at site $i$,

$$\bar{x}_{i.} = \sum_{j=1}^{N} x_{ij}/N;$$

the average deviation for individual $j$,

$$\bar{x}_{.j} = \sum_{i=1}^{M} x_{ij}/M;$$

and the total average deviation

$$\bar{x} = \sum_{i=1}^{M} \bar{x}_{i.}/M = \sum_{j=1}^{N} \bar{x}_{.j}/N.$$

We wish to test the null-model that deviations are due to independent mutation in each cell, mutations that occur with site- and individual-specific probabilities. For site $i$ in individual $j$, reads were simulated by drawing from a binomial distribution with parameters $n_{ij}$ and $p_{ij} = \bar{x}_{i.} + \bar{x}_{.j}$. We then calculated the variance across individuals for each site, and compared simulation results with data. If there were large sectors of epimutations in some individuals (*i.e.*, non-independence of states within individuals), the between-individual variance should be inflated. We observe the opposite for losses, whereas gains are broadly consistent with the null model.

## QTL mapping and variance partitioning

We performed linkage mapping using the R/qtl package [53]. We use both simple interval mapping (using the 'scanone' function) and composite interval mapping (using the 'cim' function) via Haley-Knott regression. We included growth temperature as a cofactor when performing linkage mapping as full model. We identified QTLs having an interaction with temperature by comparing full model with the additive model. QTLs were identified by adding LOD scores across genomic regions. The significance threshold was calculated by permuting ($n = 1000$) LOD scores and performing genome-rotations to retain the LD structure.

We estimated variance explained for identified QTLs, cis-genotype, temperature, and their interactions using a linear mixed model. We used the 'lmer' function from 'lme4' package in R [56] to implement the model

$$y = G_{\mathrm{cis}} + T + \sum_i G_{\mathrm{QTL}_i} + G_{\mathrm{cis}} \times T + \sum_i G_{\mathrm{QTL}_i} \times T + \epsilon, \tag{1}$$

where $y$ is the somatic deviation at a given genomic region, $G_{\mathrm{cis}}$ is the genotype at the cis marker, $G_{\mathrm{QTL}_i}$ is the genotype at QTL marker $i$, and $T$ is the growth temperature.

## Genome-wide association mapping (GWAS)

GWAS was performed using a linear mixed model implemented using LIMIX [57]. We used the SNP matrix ($n = 3{,}916{,}814$) from the 1001 Genomes Project filtered for SNPs with minor allele frequency greater than 5% in the Swedish populations [50].

## Estimating epimutation rates

We used the average methylation across replicate individuals of each parental line to infer the methylation state of the grand-parental individual (Fig 1 and S1 Fig). Analogously, we used a weighted average across F2 individuals homozygous for each ancestry (NN or SS) to infer the methylation state of the F1 individual (separately for the N and S chromosomes, see S1 Fig).

Any transgenerational epimutations that occurred during these two generations would give rise to differences between the inferred grand-parental and F1 states, and it should be possible to use this to estimate the epimutation rate. However, such differences could also result from estimation error, and we realized that one important source of such error would be sites heterozygous for methylation in the grand-parental generation. Such sites would lead to segregating methylation among the parental replicates, and lead to random assignment of grand-parental methylation state using our 50% rule. They are expected to be extremely rare, and

indeed there is no evidence of them in Fig 1. However, when comparing the distribution of methylation levels across sites in the averaged parental individuals compared to the averaged F2 individuals, we do see an enrichment of sites with intermediate methylation in the former (S13 Fig), presumably reflecting grand-parental heterozygosity. Another potential source of error is cryptic copy number variation, which could lead to pseudo-heterozygosity and again intermediate levels of methylation [58].

In order to guard against these errors, we filtered out all sites with ambiguous methylation state in either the grand-parental or F1 generation, conservatively retaining only sites consistent with the genome-wide distributions of gains and losses (average somatic gains per site are less than 0.2 and average somatic losses per site are less than 0.35, see S6 Fig). Almost all sites removed using this approach are due to insufficient coverage in the parental generation (which does not affect the calculations of somatic gains and losses in the rest of the paper).

With this filtered data, we estimate epimutation rates by comparing the inferred methylation state of the grandparent with that of the F1. Any difference effectively means that the F1 allele must have changed state either in early development, or via an epimutation from parent to F1, or from grandparent to parent, *i.e.*, the changes reflect two generations.

We calculated epimutation rates separately for the northern and southern lineage, and also for the two F1 individuals resulting from the two directions of the reciprocal cross (S14, S15 Fig, and S5 Table). The average per-site, per-generation epimutation rates are $\sim 0.04\%$ and $\sim 0.2\%$ respectively (S5 Table), but there are several anomalies that caution against over-interpretation of these estimates. First, losses on the northern lineage are three times higher in the NN×SS direction of the cross than in the SS×NN direction, and gains on the southern lineages are two times higher in the NN×SS direction of the cross than in the SS×NN direction. Second, when filtering for ambiguous methylation the F1 generation, we detected evidence of rapid change in this generation, consistent with the action of trans-acting modifiers. Third, the overlap in mutated sites between the two crosses is orders of magnitude higher than could be expected under any model of random mutations. The far greater sharing along the northern lineage could partly be explained if we assume (records were not kept) that the same parental individual was used as mother in one direction of the cross and father in the other (this would result in sharing of half of all epimutations that occurred in the first of the two generations of the pedigree, see S15), however, reproducible changes have previously been observed in F1 individuals [38–40]. These observations provide further evidence (see also reference [36]) that a model of random epimutations is not sufficient, and suggest that further experiments are badly needed.

Finally, we calculated epimutation rates for sites differentially methylated between the parental lines (S16 Fig and S6 Table). Consistent with the patterns in somatic deviations (Fig 3), epimutation rates are much higher for these sites: ten-fold for gains and two-fold for losses.

## Supporting information

**S1 Table. Individuals sequenced.**
(PDF)

**S2 Table. Average somatic deviations in F2 individuals.**
(PDF)

**S3 Table. Deviations in NN and SS backgrounds, separately for sites that are identical vs differ between N and S.**
(PDF)

**S4 Table. Linkage mapping results for deviations in NN×SS cross.**
(PDF)

**S5 Table. Epimutation rates using data from S14 Fig.**
(PDF)

**S6 Table. Epimutation rates using data from S16 Fig.**
(PDF)

**S1 Fig. Experimental design and residual heterozygosity. (A)** Reciprocal F2 design. **(B)** The left panel shows evidence for residual heterozygosity in the parental lines in the 1001 Genomes data. The right panel shows region where different SNPs are segregating in the reciprocal F2 populations.
(TIF)

**S2 Fig. Genetic map construction for the NN×SS cross. (A)** Genetic map and markers. **(B)** Segregation distortion in the cross and genotype frequencies across the chromosome. **(C)** Number of crossovers per chromosome in the genetic map. **(D)** Pairwise recombination fraction (upper left triangle) and LOD scores for the markers.
(TIF)

**S3 Fig. Mendelian segregation for gene-body methylation. (A)** Distribution of methylated CG sites the genome in 200 kb windows, separately for 213178 sites methylated in both parents (N-1 S-1), 109868 sites methylated only in the northern parent (N-1 S-0), and 39682 sites methylated only in the southern parent (N-0 S-1). **(B)** Genotype and relative methylation levels for 6 F2 individuals along chromosome 1. Genotypes are given by colors (NN is turquoise; SS is yellow; NS is grey), relative methylation levels by black curve.
(TIF)

**S4 Fig. Somatic deviations and chromatin environment.** Average gains (top-panel) and losses (bottom-panel) for each gene (in NN background) plotted as a function of: **(A)** gene expression for NN genotype at 16°C (data from reference [3]); **(B)** ten annotated chromatin states (*cf.* Table 2 in reference [55]). Clusters CL1–CL3 represent genes in facultative heterochromatin and with Polycomb-like silencing. Clusters CL4–CL6 are genes with heterochromatic marks. Clusters CL7–CL10 are expressed genes.
(TIF)

**S5 Fig. Patterns of somatic deviations. (A)** Gains and losses separated by four contexts (CGA, CGT, CGG and CGC). **(B)** Average gains and losses across the genome at different temperatures. **(C)** Methylation levels at (gain at previously unmethylated) CG, CHG and CHH sites near a mCG gain site. **(D)** Methylation levels at CG (loss at previously methylated), CHG and CHH sites near a CG loss site.
(TIF)

**S6 Fig. Modeling somatic deviations. (A)** Distribution of average deviations per site and per individual using data from chromosome 1 NN genotypes as an example. **(B)** Distribution of the variance between individuals across sites, $\text{Var}(X_{\cdot j})$, in data and in simulations. Top row shows the distribution for all sites, bottom row only for sites that are differentially methylated between N and S.
(TIF)

**S7 Fig. Composite Interval Mapping for average deviations. (A)** Composite Interval Mapping was applied to four different gain phenotypes and four different loss phenotypes in order

to refine peaks. For each of the four major QTL identified by combining results across 500 kb windows (two for gains and two for losses, see Fig 4) deviations were averaged over regions showing QTL effect at two temperatures. **(B)** Testing for epistasis on the QTLs for somatic deviations (using the "scantwo" function in R/qtl). Two QTLs on Chr1 and Chr5 for gains and three QTLs on Chr1, Chr4 and Chr5 for losses. The bottom triangle is the LOD scores for the full model including the interaction effect, upper triangle is LOD scores for only the interaction.
(TIF)

**S8 Fig. Deviations in parental strains.** Reaction norms for average gains and losses for parental strains.
(TIF)

**S9 Fig. Genetic architecture of deviations in reciprocal cross (SS×NN). (A)** Average deviations in 500 kb windows across genome (*cf*. Fig 3). **(B)** QTL mapping for gains and losses (*cf*. Fig 4). **(C)** Variance-partitioning results (*cf*. Fig 4). **(D)** Temperature effects on average gains and losses (in NN background) for both directions. **(E)** Genotypic effects for two gain QTL and three loss QTLs (*cf*. Fig 5).
(TIF)

**S10 Fig. GWAS of deviations in Swedish *A. thaliana*. (A)** mCG allele frequencies in populations from northern and southern Sweden [3]. **(B)** Correlation between genome-wide deviations between 10°C and 16°C. **(C)** GWAS for genome-wide gains. **(D)** GWAS for genome-wide losses.
(TIF)

**S11 Fig. Bisulfite sequencing summary statistics.** Average sequencing depth and bisulfite conversion estimated through chloroplast genome across 600 F2 individuals.
(TIF)

**S12 Fig. Gene-body methylation.** %mCHG and %mCHH on annotated protein coding genes (Araport 11) in parental lines N and S. We filtered out genes having any non-CG methylation on the gene-bodies to determine the gbM genes.
(TIF)

**S13 Fig. Possibly heterozygous mCG sites in grandparents.** Histograms for methylation levels on gbM sites averaged across parental values (N in the left panel and S on the right panel), F2 individuals homozyogous for the same ancestry, and F2 individuals heterozygous for ancestry. There are far more sites with intermediate values in the parental than in the homozygous F2 data, although the former is also supposed to be homozygous.
(TIF)

**S14 Fig. Transgenerational epimutations.** The plots compare average gbM for parents with average gbM for F2 individuals homozygous for the parental ancestry across sites, separately for the two cross-directions. Only data from chromosome 5 was used as all other chromosomes showed evidence of residual heterozygosity in the southern parental line (S1 Fig).
(TIF)

**S15 Fig. Transgenerational epimutations along lines of descent.** The transgenerational epimutation from S14 Fig are shown for each line-of-descent in the cross. "Shared" refers to the number of changed sites that are shared between the directions of the cross, separately for the northern and southern ancestry. The numbers in parentheses are for the sites that are differentially methylated sites between N and S (S16 Fig).
(TIF)

**S16 Fig. Transgenerational epimutations.** Same plots as S14 Fig but only sites that differ in methylation between the parental lines were used.
(TIF)

## Acknowledgments

We are grateful to Ortrun Mittelsten Scheid, Fred Berger, and Kelly Swarts for discussions throughout the project, and the Nordborg lab, especially Yoav Voichek, Haijun Liu and Thomas Ellis for their helpful comments on the manuscipt. We thank Bob Schmitz and Daniel Zilberman for comments on the manuscript and Rahul Pisupati's thesis. Bisulfite sequencing was performed by the Next Generation Sequencing Facility at the Vienna BioCenter Core Facilities (VBCF), a member of the Vienna BioCenter (VBC), Austria.

## Author Contributions

**Conceptualization:** Rahul Pisupati, Magnus Nordborg.

**Data curation:** Rahul Pisupati.

**Formal analysis:** Rahul Pisupati, Magnus Nordborg.

**Funding acquisition:** Magnus Nordborg.

**Investigation:** Rahul Pisupati, Viktoria Nizhynska, Almudena Mollá Morales.

**Methodology:** Rahul Pisupati, Viktoria Nizhynska, Almudena Mollá Morales.

**Project administration:** Magnus Nordborg.

**Validation:** Rahul Pisupati.

**Visualization:** Rahul Pisupati.

**Writing – original draft:** Rahul Pisupati, Magnus Nordborg.

**Writing – review & editing:** Rahul Pisupati, Magnus Nordborg.

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
