## [Decision Letter · Decision Letter 0]

9 Jan 2023

Dear Dr Nordborg,

Thank you very much for submitting your Research Article entitled 'On the Causes of Gene-Body Methylation Variation in Arabidopsis thaliana' to PLOS Genetics.

The manuscript was fully evaluated at the editorial level and by independent peer reviewers. The reviewers appreciated the attention to an important topic but identified some concerns that we ask you address in a revised manuscript. Each of the reviewers makes very useful comments that could help to improve the manuscript.

We therefore ask you to modify the manuscript according to the review recommendations. Your revisions should address the specific points made by each reviewer.

Yours sincerely,

Nathan M. Springer

Academic Editor

PLOS Genetics

John Greally

Section Editor

PLOS Genetics

Reviewer's Responses to Questions

**Comments to the Authors:**

Reviewer #1: This study details a unique genetic approach to try and identify genetic and environmental

factors underlying variation in DNA methylation in the model plant Arabidopsis thaliana. Two

distinct Swedish populations were used in a genetic cross due to their contrasting patterns and

levels of gene body DNA methylation and transposon methylation. Whole genome bisulfite

sequencing of hundreds of the resulting F2 progeny were used to test the impact of genetic

and/or the environment.

This research showed convincingly that variation in methylation that originates in the

populations is largely inherited in cis, as the variation observed mostly arose from loci that were

distinct between the two original parents used in the cross and Mendelian segregation was

observed. Changes in DNA methylation within gene bodies was almost exclusively limited to

single symmetrical sites and the rates of loss was substantially higher that the rates of gains.

Curiously gains were higher near pericentromeric regions of the genome. The rates of gains or

losses didn’t vary with temperature.

These variations were used in a linkage mapping approach to identify potentially causal cis

versus trans genetic factors influencing this molecular trait. Significant trans-genetic

associations for both gains and losses were observed, which is very exciting. Identifying the

causal nature of these QTL will be important to further understand how variation in DNA

methylation within gene bodies is shaped naturally between populations.

Overall, this is a nice study that takes advantage of a large cohort of F2 individuals to test and map variation in gbM between contrasting patterns in the parents. The comments below are intended to further improve this study.

1. In general I find Figure 3C difficult to interpret as presented.

2. It is reported that H2A.Z and variations in methylation are correlated. Please show a metagene plot of gains and losses of epimutations across gene bodies. H2A.Z is specifically localized to the 5’ end of expressed genes, so if there is a direct effect the expectation would be that methylation variation is occurring at the same location.

3. It is stated that no trans QTL are shared between gains/losses, but isn’t there one on chr5?

4. How are sequencing errors, bisulfite conversion errors and other potential sources of errors impacting the interpretation of gains and losses given these are not directly modeled in this study?

5. I would consider expanding the discussion on the differences between loss and gain rates if space allows it.

6. Figure 6b would be better presented if it was converted to a density plot. It is difficult to know where most of the data points occur in these plots. Figure 7a should also be converted to a density plot.

7. I would recommend enlarging the text in most of the figures.

Reviewer #2: The authors study CG gbM methylation changes in F2 populations obtained from a bi-parental cross between northern (NN) and southern Swedish (SS) accessions. They do reciprocal crosses to investigate parent of origin effects. The F2 individuals are obtained from seeds of a single F1(NS) individual. As a “control” they propagate the grand parental (NN or SS) lines by selfing for two generations (to match the generation of the F2 intercross). They use this design to try to approximate “epimutation rates” and compare them across N and S haplotypes. Also, using somatic deviations in CG methylation relative to the “ancestral state” as a molecular phenotype they perform QTL mapping in the F2 and identify genetic loci that contribute to somatic gains and losses of CG methylation, both in cis and in trans. One potentially interesting insight is that genetic variation leads to gbM variation via changing the epimutations rate.

The authors provide a substantial amount of new data, and it seems the bioinformatic / genetic analyses are generally carefully done. However, their results have alternative explanations that their current experimental design cannot rule out. Based on several pieces of evidence I think the authors are looking at a mixture of at least two processes that shape their results. First, a “classical” process of spontaneous epimutations as it has been studied in MA lines; and second, F1 methylome remodeling via TcM and TcdM events. My reading of their results indicate that their estimates of epimutation rates as well as their study of somatic deviations in the F2 are substantially influenced by the latter process If the latter processes contributes to their observations, their conclusions that genetic effects on epimutations rates shape natural variation in gbM in the long-term would be wrong as they F1 remodeling explanation is transient (see literature on F1 DNA methylation studies and paramutation). I think the authors could have addressed this alternative explanation with two additional modifications to their experimental design: 1. Take direct measurements on F1 hybrids (possibly even the one that produced seeds for the F2); and 2. generated more parental lines by selfing as controls, so that “classical” epimutational process could be directly contrasted within selected genomic regions (e.g. haplotypes under the cis QTL peaks). Without this, the interpretations/conclusions presented are questionable. I also think the authors could have done a better job explaining in more detail what they did. It is not straightforward following their analysis.

Below are my comments:

- The section on estimating epimutation rates should clarify it if this is based on just gbM genes and CG context.

- To estimate epimutation rates they used consensus calls in N parental replicates, and looked at consensus calls in NN homozygote F2, which would reflect the F1 state. Any difference they would see in the predicted F1 relative to the grand parental state (inferred from consensus parental calls) would be an epimutation. Any trans-generational epimutations that occurred during these two generations would give rise to differences between the inferred grand-parental and F1 states, and it should be possible to use this to estimate the epimutation rate. This strategy therefore also picks up TcM and TcdM events that occurred in the F1 hybrids that are (partly) carried over to the F2. For paramutation this is well established. So, these hybrid events are factored into their estimates. There are several anomalies and observations that tell me that the later processes contribute substantially to their detected “epimutations”.

1. Contrary to what the authors claim in the paper, their estimated rates do not look much like those for trans-generational epimutations detected with MA lines: they are much higher than the ones reported in the literature in studies of MA lines, the ratio of the rates is very different (TcdM events are most frequently observed); and – importantly- they have nearly opposite distributions across the genome compared to trans-generational epimutations (e.g. their gains peak in pericentromeric regions).

2. They observe parent of origin effects, which has also been reported to affect TcdM and TcM events.

3. They found evidence for “rapid changes in the F1 generation” during their filtering of sites, consistent with CG methylation changes originating from F1 methylome remodeling.

4. The overlap between epimutated sites between crosses is orders of magnitude higher than could be expected under any model of random epimutations (consistent with TcM and TcdM being largely systematic / reproducible processes).

5. Epimutation rates were higher for sites where the NN and SS parents were differentially methylated (TcM and TcdM events are proportionally more likely to occur in DMRs or SMPs; but I agree could be also due to the sites being more epimutable; we just don’t know).

- While TcdM and TcM events in F1 hybrids are also relatively rare (based on the literature), they are still much more frequent than what would be expected under a spontaneous epimutation model (given the known rates), which could account for the large rate differences relative to the literature. Finally, TcM events (but also TcdM events) are sRNA mediated. While a strict definition of gbM genes will unlikely have sRNA targets, I have noticed that the authors work with a list of >10K classified gbM genes (a strict list would be around 5K). Their liberal list likely includes genes with some non-CG methylation (perhaps in their introns), which likely harbor sRNA targets (even if this is just a fraction of all genes). The fact that most of their gain and loss events occur in pericentromeric regions would be further consistent with a Tcdm and TcM process as being the major driver behind these epimutation rates they get (note that the gain rate and loss rate are not highest in pericentromeric regions for trans-generations spontaneous epimutations). The sRNA argument could be bested using known sRNA target sites or by direct measurements.

- I think the possibility that we are dealing with many Tcdm or TcM induced methylation changes in the F1 potentially affects their conclusions about somatic deviations. There are at least two considerations: First, although the F2s come from a single F1, the F1 gametes constitute a population of different haplotypes. TcdM and TcM events can continue in the germline, or – more likely – be partly “corrected” in the germline. The correction is likely stochastic as permutation in A. thaliana has shown that F2 and F3 individuals only partly maintain F1 induced methylation changes (some have it while others don’t). The “correction” of such F1 events could also continue during F2 development, thus leading to somatic deviations. That these deviations are mappable could then relate to the molecular pathways that are involved in the correction process. The fact that many non-CG genes are under the detected QTL peaks would suggest to me a possible involvement of RdDM. Note that this could also explain the detected trans-effects as those trans effects are also potentially active in the gametes. Again, I think all of this could be tested with sRNA data or – as I mentioned above – with more data at F1 and parental level. Such data would have allowed the authors to delineate how much of the CG methylation changes are just a byproduct of having passed through an intra-specific F1 hybrid.

- In their analysis of somatic deviations it should be clarified if the analysis is restricted only to CG gbM methylation If so, does the window approach and the plotting of gain/losses normalized by gbm density?

- The authors use a binomial model to test the hypotheses that CG methylation changes occur in a site- and individual independent manner. For gains they cannot reject this model, while for losses they can. For gains, how can this be reconciled with their other finding that gain CG “epimutations” are autocorrelated and – by extension – with the model proposed by Brifa et al (2022, BioRxiv)? These two results seem to be contradictory. The strong binomial rejection for losses is consistent – again – with a systematic TcdM correct process (TcdM events are more frequent in many of the F1 methylation studies).

- That trans-generational epimutations occur in clusters (or as DMRs) has been shown by Denkena et al. (2021, Heredity). This study should be cited.

- In the QTL analysis, it is unclear if the 500 kb windows only focused on gbM genes.

- While there are lots of reasons why the GWAS could not replicate the F2 QTL, one interpretation is – again - that process that leads to long-term stable methylation differences between accessions (here NN and SS lines) in nature is different from the one that pick up for somatic deviations in their F2 mapping.

- For the cis QTL it would have been useful to have data on multiple parental lines at G2. This would have yielded direct comparisons of the epimutation accumulation in selfed lines on haplotypes under the cis QTL peaks and those haplotypes when passing through the F1 and/or the impact of trans-effects.

- The authors write: “. It seems more likely that the effects are a consequence of the epigenetic differences that we know exist. As mentioned previously, deviations are associated with the underlying chromatin state (Fig 3), suggesting the local epigenetic state influence them. “

Where do the authors think the chromatin state differences are coming from? If not genetic?

- How many CG sites are expected to be found in 30bp? How does this result relate to the autocorrelation epimutation model proposed by Brifa et al (2022, BioRxiv)?

Line 12: mCHG → mCG

Line 19 and 22: Quantify the range of gbM (e.g. from as low as to as high as….). This would put things into perspective. One can map QTL for phenotypes that vary very little, provided the variance is low enough. The differences are mentioned in the methods. Should be brought into the intro, for my taste.

Lines 37-39: This does not follow from this argument. For example, being fixed for slightly deleterious alleles in DNA methyltransferases will lead to bias epimutation ratio, shift steady state methylation (lead to mean differences) and should therefore be mappable. In selfing species you are sitting in a “fixed” genetic background for long enough.

Lines 44-47: Clarify sentence...can map it despite stability requires a slightly more explanation why this should be so.

Line 79: “massive”; reword

Line 162: a couple of percent; be more precise.

Reviewer #3: In their study, Pisupati and colleagues investigate what determines the variation of gene-body DNA methylation, which has been known to differ between natural accessions of A. thaliana. They use an F2 population derived from a cross of a northern and a southern Swedish genotype with high and low gbM, respectively. The authors find that somatic deviations occur mostly at sites that differ in terms of DNA methylation between the parents, and that gains and losses of methylation follow very different patterns. In a QTL mapping approach, the authors identify different loci associated with either gains or losses, most in cis but some in trans, although they cannot identify, let along validate, candidates.

The study takes an interesting angle at the question of what causes DNA methylation variation and at which rate this kind of variation arises. It aligns well with several recent studies, mostly by the Gaut, Schmitz and Johannes labs. My biggest criticism is therefore that the authors missed the opportunity to contrast their results in more detail to the findings from some of these studies.

Apart from that, the manuscript is clear and concise and nicely complements the existing body of literature on the topic. I have listed below some points where information was missing or where statements were unclear.

Minor comments:

- Unless I missed it, it seems that the authors did not provide a summary of sequencing stats. It would be informative to see the per-sample average coverage and, more importantly, the bisulfite conversion rates. Differences in conversion rates could potentially affect methylation calls. Given the number of individual samples, I doubt that the results would be seriously affected, but supplying these data should be part of the standard material supplied with this kind of study.

- The title: Do the authors really provide insights into the “cause” of DNA methylation variation? The manuscript does not even speculate on potential candidate genes in the QTL mapping intervals that could explain the variation, and it surely does not provide an explanation. I therefore find the title somewhat misleading.

- Figure 2 left me a bit mystified. I think it would be helpful to explain the symbols, colors, etc. in the legend. Particularly in the lower panel, the numbers and the varying number of rows of “methylation calls” are not self-explanatory.

- Line 136: effecting -> affecting

- Line 155: “the 500 kb size was empirically determined” -> How? If I am not mistaken, this is mentioned nowhere in the text.

- Fig S1: what is the purpose of the “uncrossed” and seemingly un-sequenced individuals in the 2nd row from the top (outer left and outer right)?

- Fig S3: this may be due to the way the data is presented here, but at least in these graphs, the differences between the genotypes seem drastic. From the legend, it appears that this figure shows ALL mCG sites, not only the gbM ones. Is that the case? If so, it would appear that SS is almost completely depleted of overall CG methylation!?

- Fig 3D: “DMC” is not explained anywhere, nor does it appear anywhere in the text.

**Have all data underlying the figures and results presented in the manuscript been provided?**

Reviewer #1: Yes

Reviewer #2: **No: **Did not check if data had been deposited

Reviewer #3: Yes

PLOS authors have the option to publish the peer review history of their article (what does this mean?). If published, this will include your full peer review and any attached files.

Reviewer #1: No

Reviewer #2: No

Reviewer #3: No

---

## [Decision Letter · Decision Letter 1]

31 Mar 2023

Hello Magnus,

We are pleased to inform you that your manuscript entitled "On the Causes of Gene-Body Methylation Variation in Arabidopsis thaliana" has been editorially accepted for publication in PLOS Genetics. Congratulations!  Thank you for the careful and comprehensive responses to the reviewer comments - we really appreciate the revisions and responses.

Yours sincerely,

Nathan M. Springer

Academic Editor

PLOS Genetics

John Greally

Section Editor

PLOS Genetics

Comments from the reviewers (if applicable):

Reviewer's Responses to Questions

**Comments to the Authors:**

Reviewer #2: I have read the authors' responses to my review and their implementations. The authors have addressed my comments and I have no more concerns.

Reviewer #3: The authors have substantially improved their manuscript by including more elaborate discussions of recent literature. In general, the authors have addressed all comments I had made in the initial review; I have no further comments.

**Have all data underlying the figures and results presented in the manuscript been provided?**

Reviewer #2: **No: **Not sure

Reviewer #3: Yes

PLOS authors have the option to publish the peer review history of their article (what does this mean?). If published, this will include your full peer review and any attached files.

Reviewer #2: No

Reviewer #3: No

**Data Deposition**

http://datadryad.org/submit?journalID=pgenetics&manu=PGENETICS-D-22-01391R1

**Press Queries**

---

## [Editor Report · Acceptance letter]

28 Apr 2023

PGENETICS-D-22-01391R1 

On the Causes of Gene-Body Methylation Variation in Arabidopsis thaliana 

Dear Dr Nordborg, 

We are pleased to inform you that your manuscript entitled "On the Causes of Gene-Body Methylation Variation in Arabidopsis thaliana" has been formally accepted for publication in PLOS Genetics! Your manuscript is now with our production department and you will be notified of the publication date in due course.

With kind regards,

Timea Kemeri-Szekernyes

PLOS Genetics

On behalf of:
